# A Multifaceted Look at Starlink Performance

## ABSTRACT

In recent years, Low-Earth Orbit (LEO) mega-constellations have emerged as a promising network technology and have ushered in a new era for democratizing Internet access. The Starlink network from SpaceX stands out as the only consumer-facing LEO network with over 2M+ customers and more than 4000 operational satellites. In this paper, we conduct the first-of-its-kind extensive multi-faceted analysis of Starlink network performance leveraging several measurement sources. First, based on 19.2M crowdsourced M-Lab speed test measurements from 34 countries since 2021, we analyze Starlink global performance relative to terrestrial cellular networks. Second, we examine Starlink's ability to support real-time web-based latency and bandwidth-critical applications by analyzing the performance of (i) Zoom video conferencing, and (ii) Luna cloud gaming, comparing it to 5G and terrestrial fiber. Third, we orchestrate targeted measurements from Starlink-enabled RIPE Atlas probes to shed light on the last-mile Starlink access and other factors affecting its performance globally. Finally, we conduct controlled experiments from Starlink dishes in two countries and analyze the impact of globally synchronized "15-second reconfiguration intervals" of the links that cause substantial latency and throughput variations. Our unique analysis provides revealing insights on global Starlink functionality and paints the most comprehensive picture of the LEO network's operation to date.

## 1 INTRODUCTION

Over the past two decades, the Internet's reach has grown rapidly, driven by innovations and investments in wireless access [23, 44, 45] (both cellular and WiFi) and fiber backhaul deployment that has interconnected the globe [3, 9, 11, 25, 72]. Yet, the emergence of Low-Earth Orbit (LEO) satellite networking, spearheaded by ventures like Starlink [62], OneWeb [47], and Kuiper [4], is poised to revolutionize global connectivity. LEO networks consist of mega-constellations with thousands of satellites orbiting at 300–2000 km altitudes, offering ubiquitous *low latency* coverage worldwide. Consequently, these networks are morphing into "global ISPs" capable of challenging existing Internet monopolies [63], bridging connectivity gaps in remote regions [37, 65], and providing support in disaster-struck regions with impaired terrestrial infrastructure [22].

*Starlink* from SpaceX stands out with its expansive fleet of 4000 satellites catering to 2M+ subscribers across 63 countries [56, 71]. The LEO operator plans to further amplify its coverage and quality of service (QoS) by launching $\approx$ 42,000 additional satellites in the coming years [16]. However, despite significant global interest and the potential to impact the existing Internet ecosystem, only limited explorations have been made within the research community to understand Starlink's performance. The challenge stems from a lack of global vantage points required to accurately gauge the network's performance since factors such as orbital coverage, density of ground infrastructure, etc., can impact connectivity across regions. Initial studies have resorted to measurements from a handful of geographical locations [26, 36, 37, 41] or extrapolated global performance through simulations [27] and emulations [33]. However,

the community agrees on the limited scope of such studies and has made open calls to establish a global LEO measurement testbed to address this challenge [49, 57, 68]. Some researchers have navigated around this hurdle by exploring alternative measurement methods, e.g., by targeting exposed services behind user terminals [20] or by mining speed test reports shared on social media platforms, such as Reddit [67]. While innovative, we argue that these techniques are insufficient to uncover the intricacies affecting the network, specifically its capability to support web applications.

This paper addresses this knowledge gap and provides the first comprehensive multi-faceted measurement study on Starlink. Our work is distinct from previous works in several ways. Firstly, we examine the global evolution of the network since 2021 by analyzing the M-Lab speed test measurements [14] from 34 countries (largest so far). We complement our investigation through active measurements over 98 RIPE Atlas [55] probes in 21 countries and conduct high-resolution experiments over controlled terminals in two European countries to investigate real-time web application performance and factors impacting Starlink's last-mile access. Specifically, we make the following contributions.

**(1)** We present a longitudinal study of global Starlink latency and throughput performance from M-Lab users in §4. Our analysis, incorporating $\approx$ 19.2 M samples, reveals that Starlink performs competitively to terrestrial cellular networks. However, its performance varies globally due to infrastructure deployment differences, and is dependent on the density and closeness of ground stations and Point-of-Presence (PoP). We also observe signs of *bufferbloating* as Starlink's latency increases by several factors under traffic load.

**(2)** We assess and compare the performance of real-time web applications, specifically Zoom video conferencing and Amazon Luna cloud gaming, to terrestrial networks (§5). We find that, under optimal conditions, Starlink is capable of supporting such applications, matching the performance over cellular; however, we do observe some artifacts due to the network's periodic reconfigurations.

**(3)** We perform targeted measurements from Starlink RIPE Atlas [55] probes and leverage their diverse locations to characterize the satellite last-mile "bent-pipe" performance (§6.1). We find that the "bent-pipe" latency within the dense 53° shell remains consistent worldwide ($\approx$ 40 ms), and is significantly lower to yet incomplete 70° and 97.6° orbits. We also find evidence of Starlink inter-satellite links (ISLs) connecting remote regions, showcasing superior performance to terrestrial paths in our case study.

**(4)** Our high-frequency measurements from terminals in two European countries confirm that Starlink performs network reconfigurations every 15s, leading to noticeable latency and throughput degradations at sub-second granularity. By correlating data from our terminals, one covered by 53° and the other restricted to 70° and 97.6° connectivity, we find that the reconfigurations are globally synchronized events and likely independent of satellite handovers.

Leveraging multi-dimensional, global, and controlled high resolution measurements, our findings distinctively advance the state-of-the-art by illuminating Starlink's global performance and the

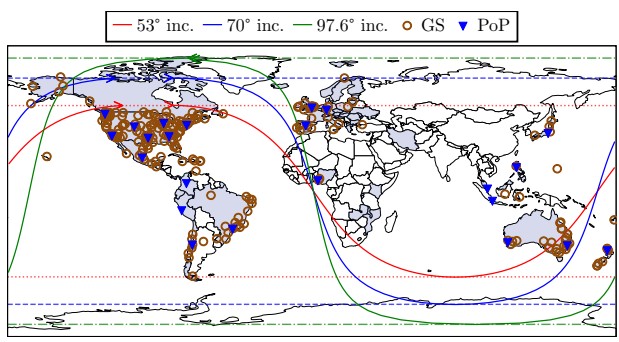

**Figure 1: Orbits of three Starlink inclinations and crowd-sourced Ground Station (GS) and Point-of-Presence (PoP) locations [48]. Shaded regions depict Starlink's service area.**

influence of internal network operations on real-time web applications. To foster reproducibility, we plan to publish our collected dataset and (measurement and analysis) scripts upon acceptance.

## 2 BACKGROUND

Starlink is a LEO satellite network operated by SpaceX that aims to provide global Internet coverage through a fleet of satellites flying at ≈ 500 km above the Earth's surface. The majority of Starlink's operational 4000 satellites lie within the 53° shell, which only covers parts of the globe (see Figure 1). The 70° and 97.6° orbits allow serving regions near the poles. These other shells however have fewer satellites (see Appendix A, Table 2 for constellation details).

Figure 2 shows the cross-section of Starlink end-to-end connectivity. To access the Internet over the Starlink network, end-users require a dish, a.k.a. "Dishy"[1], that communicates with satellites visible above 25° of elevation through phased-array antennas using Ku-band (shown as User Link (UL)). Starlink satellites, equipped with multiple antennas subdivided into beams, can connect to multiple terminals simultaneously [19] and relay all connections to a ground station (GS) on a Ka-band link (shown in green). The connection forms a direct "bent-pipe" in case the terminal and GS lie within a single satellite's coverage cone; otherwise, the satellites can relay within space to reach far-off GSs via laser inter-satellite links (ISLs), forming an "extended bent-pipe". Note that not all Starlink satellites are ISL-capable and it is difficult to effectively estimate ISL usage as Starlink satellites have no user visibility at IP layer and, therefore, do not show up in `traceroutes`.

Finally, the GSs relay traffic from satellites to Starlink point-of-presence (PoP) through a wired connection, which routes it to the destination server via terrestrial Internet [7]. The public availability of GS deployment information differs across countries. No official source exists, so we rely on crowdsourced data for the geolocations of GSs and PoPs [48], which is also shown in Figure 1.

## 3 MEASUREMENT METHODOLOGY

### 3.1 Global Measurements

***Measurement Lab (M-Lab)*** M-Lab [14] is an open-source project that allows users to perform end-to-end throughput and latency speed tests from their devices to 500+ servers in 60+ metropolitan areas [32]. Google offers M-Lab measurements when a user

---

1

[1] We use "Dishy" and "user terminal" interchangeably in the paper.

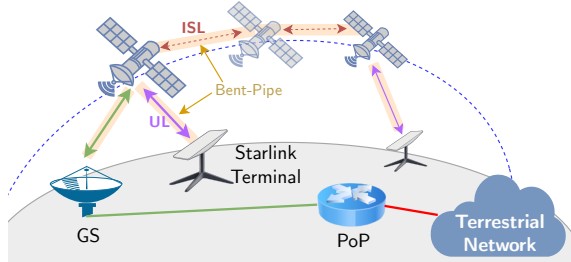

**Figure 2: Starlink follows "bent-pipe" connectivity as traffic traverses the client-side terminal, one or more satellites via inter-sat links (ISLs), nearest ground station (GS), ingressing with the terrestrial Internet via a point-of-presence (PoP).**

searches for "speed test" [31], serving as the primary source of measurement initiations [14, 21, 51]. At its core, M-Lab uses the Network Diagnostic Tool (NDT) [40], which measures uplink and downlink performance using a single 10 s WebSocket TCP connection. The platform also records fine-grained transport-level metrics (`tcp_info`), including goodput, round-trip time (RTT) and losses, along with IP, Autonomous System Number (ASN), and geolocation of both the end-user device and the selected M-Lab server. We identify measurements from the Starlink clients via their ASN (AS14593). The M-Lab dataset includes samples from 59 out of 63 countries where Starlink is operational. We restrict our analysis to ndt7 measurements, which use TCP BBR and countries with *at least 1000 measurements*, resulting in 19.2 M M-Lab measurement samples from 34 countries. Our analysis chronicles the global Starlink operation from its inception, as the first measurement samples in our dataset are dated to June 2021, which is closely aligned with the launch of Starlink v1.0 and v1.5 satellites [29]. We find that the M-Lab server selection algorithm assigns the geographically closest server to the estimated client location [30], which might not always be optimal for Starlink, given its PoP-centered architecture. While we examine such artifacts by contrasting the M-Lab and RIPE Atlas results (§6.1), we approached our analysis with caution, particularly when examining fine-grained region-specific insights.

***RIPE Atlas.*** RIPE Atlas is a measurement platform that the networking research community commonly employs for conducting measurements [55]. The platform comprises thousands of hardware and software probes scattered globally, enabling users to carry out active network measurements such as `ping`, `traceroute`, and DNS resolution to their chosen endpoints. In our study, we utilized 98 Starlink RIPE Atlas probes across 21 countries (see Figure 3). Our measurement targets were 145 data centers from *seven* major cloud providers – Amazon EC2, Google, Microsoft, Digital Ocean, Alibaba, Amazon Lightsail, and Oracle (see Appendix B). The chosen operators represent the global cloud market [3, 25, 34, 72] and ensure that our endpoints are close to Starlink PoPs, which are usually co-located with Internet eXchange Point (IXP) or data center facilities [20, 26]. We perform ICMP `traceroutes` from Atlas probes to endpoints situated on the same or neighboring continent. We extract and track per-hop latencies between Starlink probe terminal-to-GS (identified by static `100.64.0.1` address), GS-to-PoP (`172.16/12` address) and PoP-to-endpoint at 2 s intervals [49]. Additionally, to improve PoP geolocations, we extract semantic location embeddings in reverse DNS PTR entry,


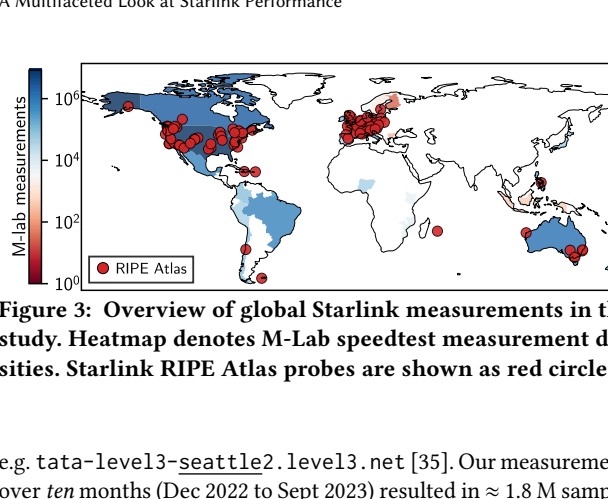

Figure 3: Overview of global Starlink measurements in this study. Heatmap denotes M-Lab speedtest measurement densities. Starlink RIPE Atlas probes are shown as red circles.

e.g. `tata-level3-`_`seattle`_`2.level3.net` [35]. Our measurements over *ten* months (Dec 2022 to Sept 2023) resulted in $\approx 1.8$ M samples.

## 3.2 Real-time Web Application Measurements

***Zoom Video Conferencing*** We experimented with Zoom videoconferencing [74] due to its popularity in the Internet ecosystem [12] as well as latency and bandwidth-critical operational requirements. We set up a call between two parties, one using a server with access to an unobstructed Starlink dish and high-speed terrestrial fiber over 1 Gbps Ethernet. The other end was on an AWS machine located close to the assigned Starlink PoP. We set up virtual cameras and microphones on both machines, which were fed by a pre-recorded video of a person talking, resulting in bidirectional transmission. Both machines were time-synchronized to local stratum-1 NTP servers and we recorded (and analyzed) Zoom QoS leveraging the open-source toolchain from [42] that yields sub-second metrics.

***Cloud Gaming.*** We also experiment with cloud gaming due to its demanding high throughput and low delay requirements [43]. We leverage the automated system by Iqbal et al. [18] to evaluate the performance of playing the racing game "The Crew" on the Amazon Luna [2] platform. The measurements are based on a customized streaming client that records end-to-end information about media streams, such as frame and bitrate. The system also utilizes a bot that executes in-game actions at pre-defined intervals that trigger a predictable and immediate visual response. In post-processing, their analysis system detects the visual response and computes the *game delay* as the time passed since the input action was triggered. Amazon Luna serves games at a resolution of up to 1920×1080 at 60 FPS and adaptively reduces the resolution to, e.g., 1280×720. We ran the game streaming client on the same machine as the Zoom measurements, additionally setting up a 5G modem to compare Starlink against cellular network. Similar to Zoom, the Luna game server was on AWS server close to our Starlink PoP ($\approx 1$ ms RTT).

## 3.3 Targeted Measurements

A significant limitation of our global measurements is their lack of sub-second visibility, which is essential for understanding the intricacies of Starlink network behavior. To allow us to obtain microscopic understanding, we orchestrated a set of precise, tailored, and controlled experiments, utilizing two Starlink terminals as vantage points (VPs) situated in two European countries. One connects to the 53° shell while the other, deployed in a high latitude location, can be shielded to confine its communication to the 70° and 97.6° orbits

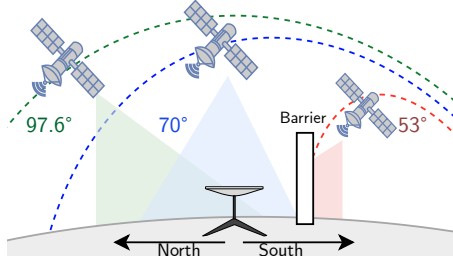

Figure 4: Field-of-view experiment setup. Dishy, deployed at a high latitude location, is obstructed by a metal shielding, which restricts its connectivity to the 70° and 97.6° orbits.

(see Figure 4). We placed a metal sheeting[2] barrier at the South-facing angle of the terminal, which obstructed its view from the 53° inclinations. We verify with external satellite trackers [28, 54] that the terminal only received connectivity from satellites in 97° or 70° inclinations, which resulted in brief *connectivity windows* followed by periods of no service. We performed experiments using the Isochronous Round-Trip Tester (`irtt`) [52] and iperf [17] tools. The `irtt` setup records RTTs at high resolutions (3 ms interval) by transmitting small UDP packets. The `irtt` servers were deployed on cloud VMs in close proximity to the assigned Starlink PoP of both VPs (within 1 ms) – minimizing the influence of terrestrial path on our measurements. We used `iperf` to measure both uplink and downlink throughput and record performance at 100 ms granularity. Simultaneously, we polled the gRPC service on each terminal [61] every second to obtain the connection status information.

## 4 GLOBAL STARLINK PERFORMANCE

We use the minimum RTT (minRTT) reported during ndt7 tests to the closest M-Lab server globally to quantify the baseline network performance. This metric is not affected by queuing delays prevalent during throughput measurements which results in elevated latencies. To put the Starlink latency into context, we select speedtests originating from terrestrial serving-ISPs to capture mobile network traffic. We filter measurements from devices connected to the top-3 mobile network operators (MNOs) in each country (see Appendix C for details). Note that our criterion results in a mix of wired and wireless access networks since M-Lab does not provide a way to distinguish between the two. Our endpoint selection remains the same for both Starlink and terrestrial networks (see §3.1).

***Global View.*** Figure 5 shows that, for a majority of countries, clients using terrestrial ISPs experience better latencies over Starlink. While the median latency of Starlink hovers around 40–50 ms in most countries, this distribution varies significantly across geographical regions. For instance, in Colombia, Starlink clients report better latencies than those utilizing established terrestrial networks. Conversely, in Manila (The Philippines), Starlink's performance is notably inferior (Figure 6). The uneven distribution of GSs and PoPs (Figure 1) may explain the latency differences; the USA, which experiences significantly lower latencies, also boasts a robust ground infrastructure. Similar trends are seen in Kenya and Mozambique, where the closest PoP is located in Nigeria.

---

[2]Metal sheeting was chosen due to its ability to act as a Faraday shield, blocking the RF emissions from satellites.

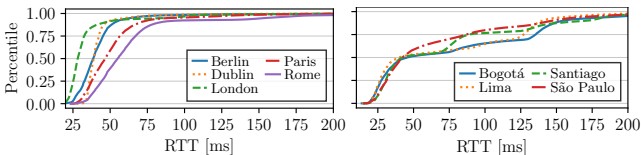

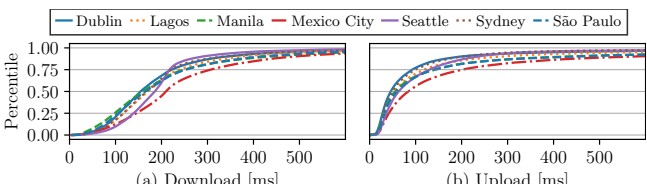

**Figure 5: Median of minimum RTT (in ms) of devices connected via Starlink (left) and top-3 serving ISPs (right) in the same country to the nearest M-Lab server.**

**Figure 6: Starlink latency distribution from select cities in each continent.**

**Figure 7: Distributions of M-Lab minRTTs from select cities in Europe and South America, respectively.**

**Figure 8: RTT inflation (`maxRTT-minRTT`) during M-Lab speedtests over Starlink: (a) download, (b) upload traffic.**

***Well-Provisioned Regions.*** Even though a significant portion of global Starlink measurement samples originate from Seattle ($\approx 10\%$), the region shows consistently low latencies, with the 75th percentile well below 50 ms (Figure 6). Contributing factors can be dense GS availability or internal service prioritization for Starlink's head-quarters. However, we observe that Starlink performance is fairly consistent across the USA, confirming that Seattle is not an anomaly but the norm (see Figure 21a in Appendix D). This result highlights the LEO network's potential to bridge Internet access disparities, which significantly affects the quality of terrestrial Internet in the USA [38, 50]. Europe is also relatively well covered with GSs but hosts only three PoPs that are in the UK, Germany, and Spain. Proximity to the nearest PoP correlates strongly with minRTT performance in Figure 7 – Dublin, London, and Berlin exhibit latencies comparable to the US, while for Rome and Paris, the 75th percentile is $\approx 20$ ms longer. Unlike US, Starlink in EU has significantly longer tail latencies, often surpassing 100 ms.

***Under-Provisioned Regions.*** Starlink's superior performance in Colombia hints at its potential for connecting under-provisioned regions. However, Figure 6 shows that Starlink in South America (SA) trails significantly behind the US and Europe, with the 75th percentile exceeding 100 ms and tail at 200 ms. We observe similar performances in Oceania (see Figure 21b in Appendix D). By extracting the share of satellite vs. terrestrial path (from PoP to M-Lab servers, see Figure 18 in Appendix D)[3], we find that the majority of SA Starlink latency comes from the bent-pipe. In contrast, latencies from Mexico and Africa (except Nigeria) show significant terrestrial influence, which we allude to non-optimal PoP assignments by Starlink routing policies.

We observed an interesting impact of ground infrastructure deployment in the Philippines, where a local PoP was deployed in May 2023. Prior to this, Starlink traffic from the Philippines was directed to the nearest Japanese PoP, traversing long submarine links to reach the geographically closest M-Lab server in-country – evident from Figure 19 in Appendix D which shows additional 50–70 ms RTT incurred by Philippine users to reach in-country vs. Japanese M-Lab servers. However, post-May 2023, the latencies to

in-country servers were reduced by 90% as the traffic was routed via the local PoP. Despite such artifacts, Starlink shows an evident trend towards more consistent sub-50 ms latencies globally over the past 17 months, specifically evident in Sydney (Figure 23a in Appendix D). We conclude that while Starlink lags behind terrestrial networks today, the gap will continue to shrink as the ground (and satellite) infrastructure expands.

***Latency Under Load.*** Recent findings suggest that Starlink may be susceptible to *bufferbloat* [15, 24], wherein latencies during traffic load can increase significantly due to excessive queue buildups [41]. To explore this globally, we evaluate the RTT inflation, i.e., the difference between the maximum and minimum RTT observed during a speed test. Figure 8 reveals significantly increased RTTs under load within Starlink globally. During active downloads (Figure 8a), the Starlink-enabled clients can experience $\approx 2$–$4\times$ increased RTTs, reaching almost 400–500 ms. While such inflations are consistent across *all* Starlink service areas, they are more prominent in regions with subpar baseline performance, e.g., Mexico. Note that the Starlink latency under load is not symmetric. The 60th percentile of RTT during uploads increases to $\leq 100$ ms globally (see Figure 8(b)) compared to $\approx 200$ ms during downloads. We observe similar behavior while conducting `iperf` over our controlled terminals. Possible explanations can be queue size differences at the client-side Starlink router (affecting uploads), the ground station (affecting downloads), or satellites (impacting both). It is also plausible that Starlink employs active queue management (AQM) techniques [1] to moderate uplink latencies under congestion. This approach, however, may adversely affect the performance of applications that demand both high bandwidth and low latency – which we explore in §5.

***Goodput.*** Figure 9 shows Starlink download and upload goodputs from speedtest globally. Unlike latencies (Figure 6), the goodput distributions appear relatively homogeneous. Most Starlink clients achieve $\approx 50$–$100$ Mbps download and $\approx 4$–$12$ Mbps upload rates at the 75th percentile. We do also not find any correlation between baseline latencies (see Figure 6) and upload/download goodput, evident from the contrasting cases of Dublin and Manila. However, we observe an inverse correlation between loss rates and good-puts; increasing from 4–8% at the 75th-percentile (see Figure 20 in Appendix D). Seattle, notable for its latency performance, records

---

[3]We subtract the latency to the Starlink PoP reported by M-Lab's reverse traceroutes from the end-to-end TCP minRTT.

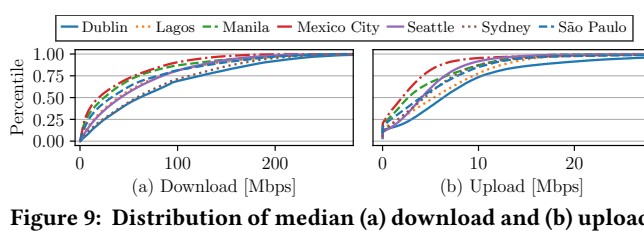

**Figure 9: Distribution of median (a) download and (b) upload goodput over Starlink from selected cities globally.**

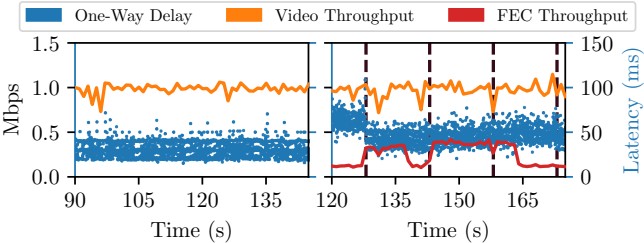

**Figure 10: Uplink Zoom video traffic over a terrestrial network (left) and Starlink (right). Vertical dashed lines show Starlink reconfiguration intervals.**

average goodputs. Given its high measurement density at this location, this trend might be attributable to Starlink's internal throttling or load-balancing policies aimed at preventing congestion on the shared network infrastructure [64]. We also find that over the past 17 months, Starlink goodputs have stabilized rather than increased, with almost all geographical regions demonstrating similar performance (shown in Figure 23 in Appendix D).

> *Takeaway #1* — Starlink exhibits competitive performance to terrestrial ISPs on a global scale, especially in regions with dense GS and PoP deployment. However, noticeable degradation is observable in regions with limited ground infrastructure. Our results further confirm that Starlink is affected by bufferbloat. Over the past 17 months, Starlink appears to be optimizing for consistent global performance, albeit with a slight reduction in goodput, likely due to the increasing subscriber base.

## 5  REAL-TIME APPLICATION PERFORMANCE

While the global Starlink performance in §4 is promising for supporting web-based applications, it does not accurately capture the potential impact of minute network changes caused by routing, satellite switches, bufferbloating, etc., on application performance. Real-time web applications are known to be sensitive to such fluctuations [8, 18, 41]. In this section, we examine the performance of Zoom and Amazon Luna cloud gaming over Starlink (see §3.2 for details). This allows us to assess the suitability of the LEO network to meet the requirements of the majority of real-time Internet-based applications, as both applications impose a strict latency control loop. Cloud gaming necessitates high downlink bandwidth, while Zoom utilizes uplink and downlink capacity simultaneously.

***Zoom Video Conferencing.*** Figure 10 shows samples from Zoom calls conducted over a high-speed terrestrial network and over Starlink. The total uplink throughput over Starlink is slightly higher, which we trace to FEC (Forward Error Correction) packets that are frequently sent in addition to raw video data (on average 146±99 Kbps vs. 2±2 Kbps over terrestrial). The frame rate, inferred from the

| | Terrestrial | Cellular | Starlink |
|---|---|---|---|
| Idle RTT (ms) | 9 | 46 | 40 |
| Throughput (Mbps) | 1000 | 150 | 220 |
| Frames-per-second | 59±1.51 | 59±1.68 | 59±1.63 |
| Bitrate (Mbps) | 23.08±0.38 | 22.82±4.24 | 22.81±2.16 |
| Time at 1080p (%) | 100 | 94.11 | 99.45 |
| Freezes (ms/min) | 0±0 | 0±220.34 | 0±119.74 |
| Inter-frame (ms) | 17±3.65 | 18±11.1 | 16±6.76 |
| Game delay (ms) | 133.53±19.79 | 165.82±23.55 | 167.13±23.12 |
| RTT (ms) | 11±13.41 | 39±17.06 | 50±16.28 |
| Jitter buffer (ms) | 15±3.27 | 12±1.33 | 15±3.35 |

**Table 1: The game metrics are aggregated over 150 minutes of playtime per connection. Values denote median±SD and the worst performer is highlighted.**

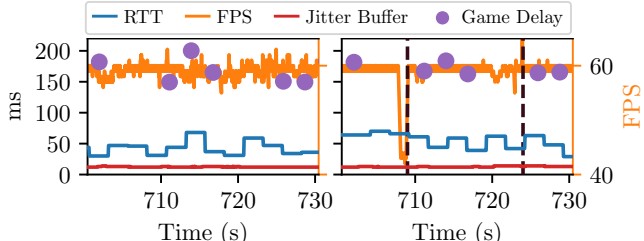

**Figure 11: Cloud gaming over 5G (left) and Starlink (right). Vertical dashed lines show Starlink reconfiguration intervals.**

packets received by the Zoom peer, does not meaningfully differ between the two networks (≈ 27 FPS). Note that, since Zoom does not saturate the available uplink and downlink capacity, it should not be impacted by bufferbloating. Yet, we observe a slightly higher loss rate over LEO, which the application combats by proactively utilizing FEC. The uplink one-way delay (OWD) over Starlink is higher and more variable compared to the terrestrial connection (on average 52±14 ms vs. 27±7 ms). All observations also apply to the downlink except that Starlink's downlink latency (35±11 ms) is similar to the terrestrial connection (32±7 ms). Our analysis broadly agrees with [73] but our packet-level insight reveals bitrate fluctuations partly caused by FEC. Further, our Starlink connection was more reliable and we did not experience second-long outages.

Interestingly, we observe that the Starlink OWD often noticeably shifts at interval points that occur at 15 s increments. Further investigation reveals the cause to be the Starlink *reconfiguration interval*, which, as reported in FCC filings [66], is the time-step at which the satellite paths are reallocated to the users. Other recent work also reports periodic link degradations at 15 s boundaries in their experiments, with RTT spikes and packet losses of several orders [26, 49, 68]. We explore the impact of reconfiguration intervals and other Starlink-internal actions on network performance in §6.

***Amazon Luna Cloud Gaming.*** Table 1 shows 150 minutes of cloud gaming performance over terrestrial, 5G cellular, and Starlink networks. Overall, all networks realized close to 60 FPS playback rate at consistently high bitrate (≈ 20 Mbps). Starlink lies in between the better-performing terrestrial and cellular in terms of bitrate fluctuations, frame drops and freezes[4]. Starlink exhibits the highest

---

[4]Freeze is when the inter-frame delay (IFD) is larger than $\max(3 \times IFD, IFD + 150)$.

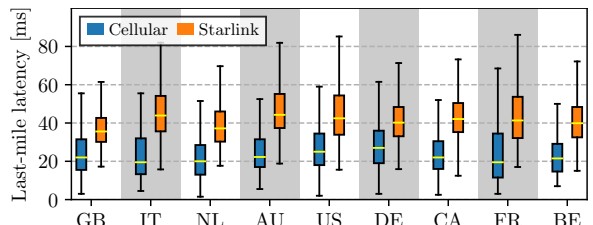

**Figure 12: Last-mile latencies for different countries. "Starlink" denotes satellite bent-pipe over RIPE Atlas while "Cellular" wireless access from Speedchecker [11].**

game delay, i.e., the delay experienced by the player between issuing a command and witnessing its effect. Specifically, the wired network delivers the visual response about 2 frames ($\approx$ 33 ms) earlier than both 5G and Starlink. While examining the gaming performance over time, we observe occasional drops to < 20 FPS over Starlink (see Figure 11), that coincide with Starlink's reconfiguration interval. These fluctuations are only visible at sub-second granularity and, hence, are not reflected in global performance analysis (§4).

Despite these variations, Starlink's performance remains competitive with 5G, highlighting its potential to deliver real-time application support, especially in regions with less mature cellular infrastructure. Note, however, that our Starlink terminal was set up without obstructions and the weather conditions during measurements were favorable to its operation [37]. Different conditions, especially mobility, may change the relative performance of Starlink and cellular, which we plan to explore further in the near future.

> *Takeaway #2* — Starlink is competitive with the current 5G deployment for supporting demanding real-time applications. We also observe that Starlink experiences regular performance changes every 15s linked to its reconfiguration interval period. While these internal black-box parameters do influence performance to a certain extent, application-specific corrective measures, like FEC, are effective in mitigating these artifacts.

## 6 DISSECTING THE BENT-PIPE

We now attempt to uncover Starlink's behind-the-scenes operations and their impact on network performance. We follow a two-pronged approach to undertake this challenge. Our longitudinal `traceroute` measurements over RIPE Atlas accurately isolate the bent-pipe (terminal-to-PoP) global performance, allowing us to correlate it with parameters like ground station deployment, satellite availability, etc. (§6.1). We then perform high-frequency, high-resolution experiments over Starlink terminals deployed in two EU countries to zoom in on bent-pipe operation and highlight traffic engineering signatures that may impact application performance (§6.2).

### 6.1 Global Bent-Pipe Performance

***Starlink vs. Cellular Last-mile*** We contrast our end-to-end M-Lab and real-time application analysis by comparing the Starlink bent-pipe latencies from RIPE Atlas `traceroutes` to cellular wireless last-mile (device-to-ISP network) access. Given the under-representation of cellular probes in RIPE Atlas, we augment our dataset with recent comprehensive measurements from Dang et al. [11], which leveraged 115,000 cellular devices over the Speedchecker platform to analyze the performance of cellular networks worldwide. Figure 12 presents a comparative analysis of both networks

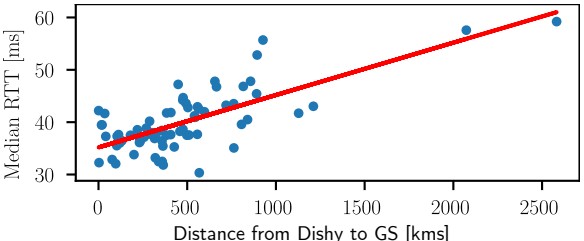

**Figure 13: Correlation between Starlink bent-pipe latency and Dishy-GS distance. Red line denotes linear regression fit.**

across countries common in both datasets. Consistent with our previous findings, we find that the Starlink bent-pipe latencies fall within 36–48 ms, with the median hovering around 40 ms for almost all countries. Similarly, we find consistent cellular last-mile latencies across all countries, but almost 1.5× less than Starlink. Recent investigations [43] report similar access latencies over WiFi and cellular networks. The bent-pipe latencies also corroborate our estimations in §4 that the terminal-PoP path is the dominant contributor to the end-to-end latency. Out of the 21 countries with Starlink-enabled RIPE Atlas probes, the only exceptions where the bent-pipe latency is significantly higher ($\approx$ 100 ms) are the Virgin Islands (US), Reunion Islands (FR), and Falkland Islands (UK). Correlating with Figure 3, we find that Starlink neither has a GS nor a PoP in these regions, which may result in traffic routing over ISLs to far-off GS leading to longer bent-pipe latencies.

***Impact of Ground Infrastructure.*** We extend our analysis by exploring the correlation between the distance from Starlink users to the GS and bent-pipe latencies. Recall that we rely on crowd-sourced data [48] for geolocating Starlink ground infrastructure since these are not officially publicly disclosed. We deduce through our `traceroutes` that Starlink directs its subscribers to the nearest GS relative to the PoP, as the GS-PoP latencies are $\approx$ 5 ms almost globally (see Figure 22 in Appendix D – sole exceptions being US and Canada with 7–8 ms, likely due to abundant availability of GSs and PoPs resulting in more complex routing). Figure 13 shows the correlation of reported bent-pipe latency with the terminal-GS distance. Each point in the plot denotes at least 1000 measurements. We observe a directly proportional relationship as bent-pipe latencies tend to increase with increasing distance to the GS. Furthermore, we find that the predominant distance between GS and the user terminal is $\leq$ 1200 km, which is also the approximate coverage area width of a single satellite from 500 km altitude [6] – suggesting that these connections are likely using direct bent-pipe, either without or with short ISL paths. Few terminals, specifically in Reunion, Falkland and the Virgin Islands, connect to GSs significantly farther away, possible only via long ISL chains, the impact of which we analyze further as a case study below.

***Case Study: Reunion Island.*** The majority of Starlink satellites (starting from v1.5 deployed in 2021) are equipped with ISLs [69], and reports from SpaceX suggest active utilization of these links [70]. Recent studies also agree with the use of ISLs [20], but point out inefficiencies in space routing [68]. Nonetheless, the invisibility of satellite hops in `traceroutes` poses a challenge in accurately assessing the latency impact of ISLs. As such, we focus on a probe in Reunion Island (RU), which connects to the Internet via Frankfurt PoP ($\approx$ 9000 km). Figure 14 segments the bent-pipe RTT between

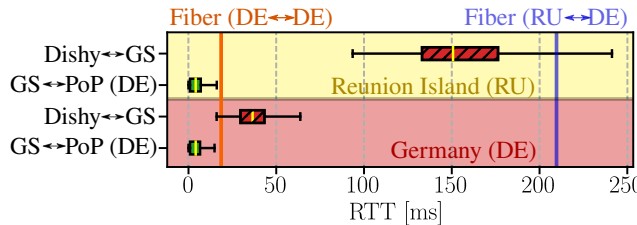

**Figure 14: Bent-pipe RTT segments from Reunion Island (yellow) vs. Germany (red) connecting to Germany PoP. Vertical lines show latency over Atlas probes connected via fiber from both locations to the Frankfurt server (PoP location).**

the user terminal (Dishy) to GS (non-terrestrial), and from GS to the PoP (terrestrial). For comparison, we also plot the RTTs from a probe within Germany (DE) connecting to the same PoP ($\approx$ 500 km, in red). The vertical lines represent the median RTT over terrestrial infrastructure from both probe locations to the PoP. Firstly, we observe minimal GS-PoP latency for both locations, verifying that the RU satellite link is using ISLs. Secondly, in RU, Starlink shows significant latency improvement over fiber ($\approx$ 60 ms). This is because the island has limited connectivity with two submarine cables routing traffic 10,000 km away, either in Asia or South America [46]. Starlink provides a better option by avoiding the terrestrial route altogether, directly connecting RU users to the dense backbone infrastructure in EU [9]. However, since the bent-pipe incurs at least 30–40 ms latency in the best-case, Starlink is less attractive in regions with robust terrestrial network infrastructure (also evident from the DE probe where fiber achieves better latencies).

***Impact of Serving Orbit.*** Recall that the majority of Starlink satellites are deployed in the 53° inclination (see Table 2 in Appendix A). Consequently, network performance for clients located outside this orbit's range may vary widely as they are serviced by fewer satellites in 70° and 97.6° orbits. Figure 15 contrasts the bent-pipe latencies of probe in Alaska (61.5685N, 149.0125W) ["A"] to probes within 53° orbit. Despite dense GS availability, the bent-pipe latencies for Alaska are significantly higher ($\approx$ 2×). The Swedish probe ["B"] at 59.6395N is at the boundary of 53° orbit but still exhibits comparable latency to Canada, UK, and Germany. Furthermore, the Alaskan probe experiences intermittent connectivity, attributed to the infrequent passing of satellite clusters within the 70° and 97.6° orbits. These findings indicate substantial discrepancies in Starlink's performance across geographical regions, which may evolve for the better as more satellites are launched in these orbits. Nevertheless, we leverage the sparse availability of satellites at the higher latitude to further dissect the bent-pipe operations in §6.2.

> *Takeaway #3* — The Starlink "bent-pipe" accounts for (on average) 40 ms of latency almost consistently globally. In certain cases where ISLs are being used, the latencies might escalate yet still outshine traditional terrestrial networks when bridging remote regions. The satellite link yields stable latencies, provided that the client is served by the dense 53° orbit.

## 6.2 Controlled Experiments

We now investigate the cause of periodic disruptions to real-time applications (§5). Specifically, we perform high-resolution measurements to gain insights into Starlink network operation.

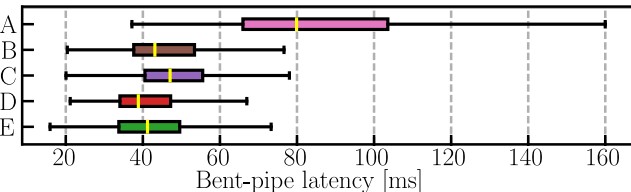

**Figure 15: Bent-pipe latencies for "A" (in Alaska) covered by the 70° and 97.6° while the rest (Sweden "B", Canada "C", UK "D", and Germany "E") are also covered by 53°.**

***Global Scheduling.*** We performed simultaneous iRTT measurements from two countries that are sufficiently geographically removed that both cannot be connected to the same serving satellite. We also verify that both terminals are assigned different PoPs located within their country. The resulting RTTs, shown in Figure 16a, vary in a consistent pattern, being comparatively stable within each Starlink reconfiguration interval but potentially changing significantly between intervals. Moreover, the time-wise alignment of reconfiguration intervals for both vantage points indicates that Starlink operates on a globally coordinated schedule, rather than on a per-Dishy or per-satellite basis. These results are in line with other recent studies [68], which also hint that Starlink utilizes a global network controller. Previous studies [13] have noticed drops in downlink throughput every 15s but have not correlated these with the reconfiguration intervals. We also observe throughput drops on both downlink and uplink, shown in Figure 16b, that occur at the reconfiguration interval boundaries. Similar to the RTT, the throughput typically remains relatively consistent within an interval, but can experience sudden changes between interval transitions. These also corroborate the periodic performance degradation in our real-time application experiments.

***Disproving Satellite Handoff Hypothesis.*** Previous works have suggested satellite or beam changes at reconfiguration interval boundaries to be the root-cause of network degradation [13, 60, 68]. To investigate this hypothesis, we deliberately obstructed the field-of-view of our high latitude Dishy to prevent it from connecting to the dense 53° orbital shell (see §3.3 for details). The restriction curtailed the number of candidate (potentially connectable) satellites to 13%. This limitation led to intermittent connectivity, characterized by brief connectivity windows with long service downtimes. By synchronizing the timings of each connectivity window with the overhead positions of candidate satellites (from CelesTrak [28] and other sources [54]), we identify several windows where the terminal can be served only by a single satellite. Figure 16c (upper) shows RTTs from one such window. The fact that there is significant RTT variance between intervals invalidates the hypothesis that the changes in RTT are caused by satellite handovers (considering a single candidate satellite during the observed period, leaving no room for hand-off occurrences). Separately, we perform the same experiment but focus on (both uplink and downlink) throughput. Similar to RTT, we also witness throughput drops at interval boundaries even when only one candidate satellite is visible.

***Scheduling Updates.*** Figure 16c (lower) shows the distribution of start and end times of the connectivity windows during our restricted field-of-view experiments. We observed a strong correlation between connectivity end times and reconfiguration interval

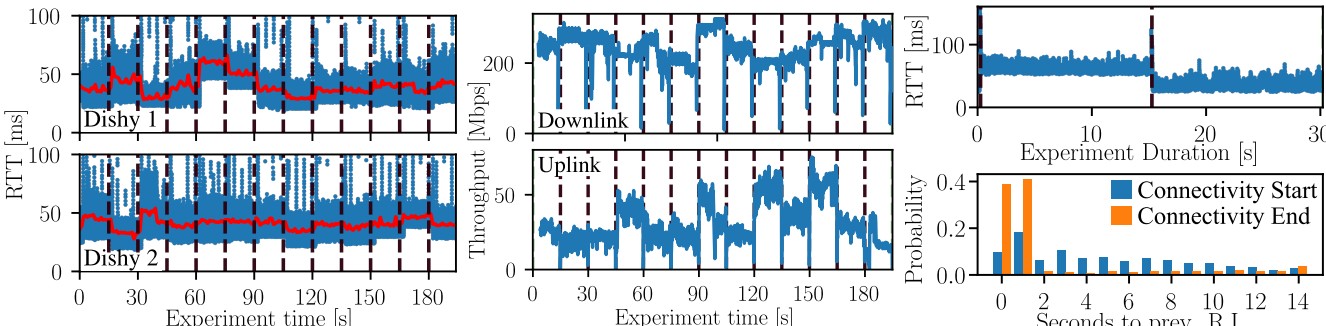

Figure 16: (left, a) iRTT latencies with Dishys in two countries connected to different ground infrastructure; (middle, b) Maximum uplink and downlink throughput over a 195-second (13 interval) period; (right, c) (upper) RTTs for a connectivity window where the Dishy was connected to only a single satellite; (lower) Probability distribution of the time between the connectivity window start / end and the previous reconfiguration interval (RI). Vertical dashed lines show Starlink reconfiguration intervals.

(RI) boundary, which is not seen with start times[5]. The result hints at internal network scheduling changes at reconfiguration interval boundaries, i.e., Starlink assigns its terminals new satellites (or frequencies) every 15s. We hypothesize that with an obstructed view, the scheduler cannot find better alternatives in the 70° and 97.6° orbits, resulting in connectivity loss at the end of the window.

*Analysis Summary.* Putting together our various observations, we theorize that Starlink relies on a global scheduler that re-allocates the user-satellite(s)-GS path every 15s. An FCC filing from Starlink implies this behavior [60] and recent studies also suggest that the LEO operator performs periodic load balancing at reconfiguration boundaries, reconnecting all active clients to satellites [20, 68]. The theory also explains our observed RTT and throughput changes when only a single candidate satellite is in view. It is plausible that Starlink may have rescheduled the terminal to the same satellite but with reallocated frequency and routing resources. Regardless, these reconfigurations result in brief sub-second connection disruptions, which may become more noticeable at the application-layer as the number of subscribers on the network increases over time.

> *Takeaway #4* — Starlink uses 15s-long reconfiguration intervals to globally schedule and manage the network. Such intervals cause latency/throughput variations at the interval boundaries. Handoffs between satellites are not the sole cause of these effects. Indeed, our findings hint at a scheduling system reallocating resources for connections once every reconfiguration interval.

## 7 RELATED WORK

LEO satellites have become a subject of extensive research in recent years, with a particular focus on advancing the performance of various systems and technologies. Starlink, the posterchild of LEO networks, continues to grow in its maturity and reach with > 2M subscribers as of September 2023 [56]. Despite its growing popularity, there has been limited exploration into measuring Starlink's performance so far. Existing studies either have a narrow scope, employing only a few vantage points [13, 36, 41] or focus on broad application-level operation [26, 73] without investigating root-causes. Ma et al. [37] embarked on a journey across Canada

with four dishes to scrutinize various factors, such as temperature and weather, that might influence Starlink's performance.

A few endeavors have attempted to unveil the operations of Starlink's black-box network. Pan et al. [49] revealed the operator's internal network topology from `traceroutes`, whereas Tanveer et al. [68] spotlighted a potential global network controller. The absence of global measurement sites poses a predominant challenge hampering a comprehensive understanding of Starlink's performance. As we show in this work, Starlink's performance varies geographically due to differing internal configurations and ground infrastructure availability. Some researchers have devised innovative methods to combat this. For example, Izhikevich et al. [20] conducted measurements towards exposed services behind the Starlink user terminal, while Taneja et al. [67] mined social media platforms like Reddit to gauge the LEO network's performance. Our study not only corroborates and extends existing findings but also stands as the most extensive examination to date. Our approach – anchored in detailed insights from 34 countries, leveraging 19.2 million crowdsourced M-Lab measurements, 2.9 million active RIPE Atlas measurements, and two controlled terminals connecting to different Starlink orbits – provides a deeper understanding of the Starlink "bent-pipe" and overall performance.

## 8 CONCLUSIONS

Despite its potential as a "global ISP" capable of challenging the state of global Internet connectivity, there have been limited performance evaluations of Starlink to date. We conducted a multifaceted investigation of Starlink, providing insights from a global perspective down to internal network operations. Globally, our analysis showed that Starlink is comparable to cellular for supporting real-time applications (in our case Zoom and Luna cloud gaming), though this varies based on proximity to ground infrastructure. Our case study shows Starlink inter-satellite connections helping remote users achieve better Internet service than terrestrial networks. However, at sub-second granularity, Starlink exhibits performance variations, likely due to periodic internal network reconfigurations at 15s intervals. We find that the reconfigurations are synchronized globally and are not caused only by satellite handovers. As such, this first-of-its-kind study is a step towards a clearer understanding of Starlink's operations and performance as it continues to evolve.

---

[5]The fact that many appear to end 1s after the boundary is an artifact of the limited (per-second) granularity of the gRPC data and that the gRPC timestamps originate from the client making the gRPC requests rather than the user terminal.

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

# A  STARLINK ORBITAL INFORMATION

| Inclination angle | # Planes | Altitude [km] | # Satellites | | |
|---|---|---|---|---|---|
| | | | In Position | Launched | Filed [58, 59] |
| 53° | 72 | 550 | 1401 | 1665 | 1584 |
| 53.2° | 72 | 540 | 1542 | 1637 | 1584 |
| 70° | 36 | 570 | 301 | 408 | 720 |
| 97.6° | 10 | 560 | 230 | 230 | 508 |

**Table 2: Starlink orbital shell design and number of operational satellites as of early October 2023 [39].**

Starlink and other emerging satellite constellations, such as OneWeb and Kuiper [4, 47], are called *megaconstellations* due to combining multiple orbital shells. Early satellite constellations for telephone services only consisted of a single shell [53]. Table 2 shows the parameters of Starlink's orbital shells. The eccentricity parameters of Starlink orbits are negligible due to the satellites' positions in Low Earth Orbit (LEO). While discussing Starlink's constellation design, we simplify the orbit into circular orbits.

# B  DATA CENTER ENDPOINTS

Table 3 shows the distribution of the data centers in our dataset by cloud provider and deployed continent. Each endpoint is a VM established in a compute-capable cloud data center location. Our selection is influenced by previous studies that have found that significant end-to-end performance differences may appear while measuring different cloud networks due to private WANs, peering agreements, etc. [10, 11]. We believe that our resulting endpoint selection reduces such biases in Internet traffic steering/routing that may affect our aggregated analysis.

# C  TOP-3 MNOS PER COUNTRY

Table 4 lists the top-3 Mobile Network Operators (MNOs) of the countries that are part of our M-Lab dataset. The selection is based on a combination of each AS's rank and the number of M-Lab measurements originating from that AS.

# D  GLOBAL STARLINK PERFORMANCE

This section of the appendix provides supporting material for the global Starlink performance analysis presented in §4. Figure 23 provides an overview over Starlink's performance over a period of one year in different cities. The plot gives an insight into the evolution of Starlink over time from a global perspective. We observe a decreasing goodput rate over time which can be attributed to an increase of Starlink users. The RTT values remain relatively stable, but high for countries which are not part of the major operational areas of Starlink. A notable exception is Sydney (AU) where the RTT decreases over time.

Figure 19 depicts the median minRTT from the Philippine tests, depending on the destination server's location. The implications are discussed in §4.

Figure 20 shows the loss rates during M-Lab download tests which are discussed in §4.

| | Data centers per continent | | | | | |
|---|---|---|---|---|---|---|
| | EU | NA | SA | AS | AF | OC |
| **Amazon EC2 (AMZN)** | 6 | 6 | 1 | 6 | 1 | 1 |
| **Google Cloud (GCP)** | 6 | 10 | 1 | 8 | - | 1 |
| **Microsoft Azure (MSFT)** | 14 | 9 | 1 | 10 | 2 | 3 |
| **Digital Ocean (DO)** | 4 | 6 | - | 1 | - | - |
| **Alibaba (BABA)** | 2 | 1 | - | 2 | - | 1 |
| **Amazon Lightsail (LTSL)** | 3 | 2 | - | 2 | - | 1 |
| **Oracle (ORCL)** | 4 | 4 | 1 | 7 | - | 2 |
| **Total** | 39 | 38 | 4 | 36 | 3 | 9 |

**Table 3: Global density of data center endpoints of different cloud providers.**

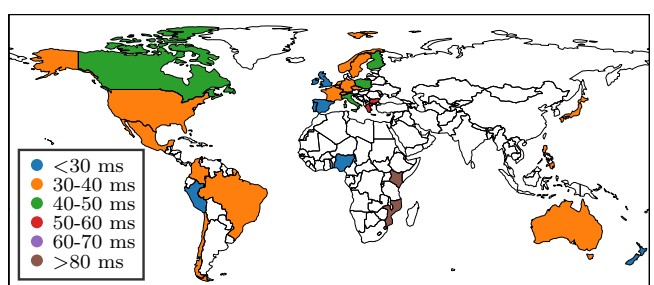

**Figure 17:  Median last-mile latency of M-lab measurement. By subtracting traceroute latency from the M-lab server to the PoP and the overall measured min RTT, we get the latency depicted in this figure**

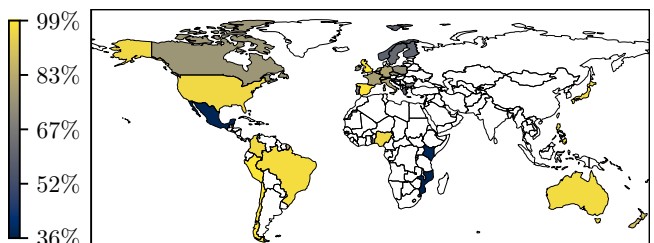

**Figure 18:  Fraction of the latency, that is estimated to be over the satellite link by dividing the latency of figure 17 with its overall latency.**

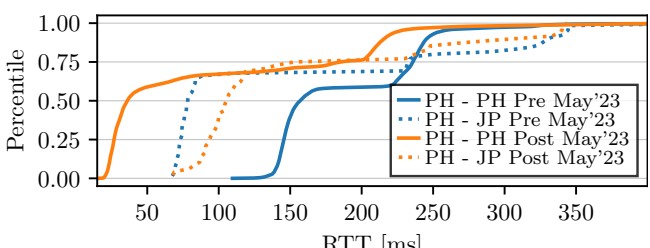

**Figure 19:  Comparing median minRTT from tests originating in Manila that targeted M-Lab servers in the Philippines and in Japan. The results are discussed in §4.**

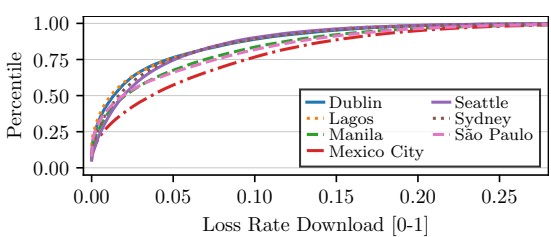

**Figure 20: The TCP packet loss rates during M-Lab download tests of selected global cities. The results are set into context in §4.**

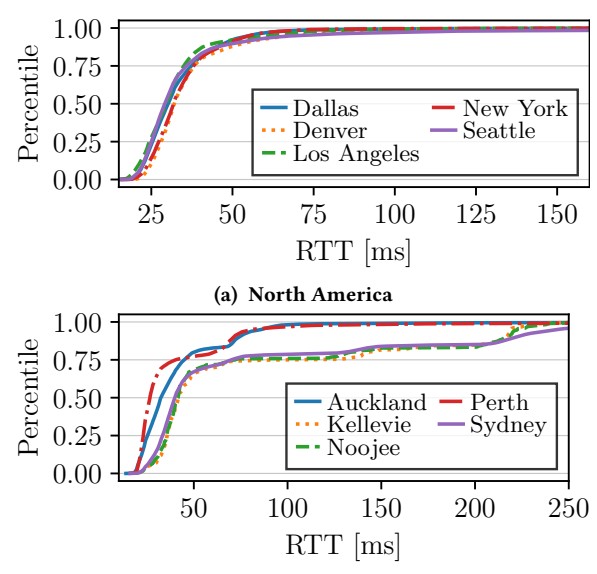

**(a) North America**

**(b) Oceania**

**Figure 21: The distribution of the minimum RTT (minRTT) during M-Lab measurements from selected cities in North America (a) and Oceania (b).**

Figure 21 shows the distribution of the minimum RTT (minRTT) during M-Lab measurements from selected cities in North America and Oceania. It complements Figure 7 that depicts selected cities in Europe and South America. Starlink's performance in North America varies only little between different cities and the latencies are low when compared globally. In Oceania, tests from Auckland and Perth exhibit a similarly low minRTT – a PoP and GSs are nearby both cities. Sydney's minRTT performance has recently (2023/06) improved, as shown in Figure 23a.

## E GLOBAL VIEW OF BENT-PIPE OPERATION

Figure 22 provides additional information about the Starlink last-mile performance analysis presented in §6.1. The figure shows the latency between Starlink ground stations (GSs) and Points of Presences (PoPs) on a world map grouped by country. It is apparent that the latencies are similar all over the world ($\leq 6$ ms) except in North America ($\geq 6$ ms). The anomaly correlates with the dense deployment of GSs and PoPs.

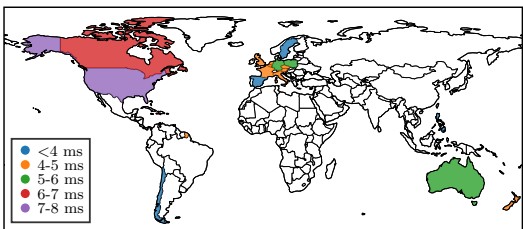

**Figure 22: Global latencies from GSs to PoPs as measured from RIPE Atlas probes.**

## F TARGETED MEASUREMENT CHALLENGES

Some of the measurements required for Section 6.2 required collection of data during time when Dishy received only patchy connectivity. We discovered that both `irtt` and `iperf` did not handle the interrupted nature of the connection well: `iperf` in particular relies on a separate TCP connection to act as the control plane. Both behaved unpredictably and unreliably on a link that has connectivity only for brief windows. Accordingly, for these experiments, we replaced `irtt` with `ping` which we set to send only a single ICMP packet with a 200 ms timeout. We then ran it in a loop.

We were unable to find a suitable replacement for `iperf`, and therefore relied upon a manual approach. When we detected the start of a connectivity window `iperf` was started. Once the connectivity window had passed, we stopped the experiment and restarted the `iperf` server. Restarting the `iperf` server was necessary to ensure that subsequent `iperf` tests could connect (`iperf3` permits only a single active connection to a server, and a loss of connectivity mid-way through an experiment can leave the server in a state where it believes an experiment is ongoing when it has in fact concluded). Automatically restarting the `iperf` server at the end of each connectivity window was not possible because the Starlink-connected computer, now without an Internet connection, could not signal to the remote `iperf` server.

An additional challenge caused by the interrupted nature of the connection was not discovered until towards the end of the targeted measurement period. The unstable connection prevented the clock on the computer connected to the Dishy from synchronising over NTP, resulting in it drifting by several seconds duration of the experiment setup. Accordingly, when the absolute timestamps of the recorded data have been analysed, they have first been adjusted to account for the time slip. The gRPC data was collected by a separate computer that did not suffer from clock drift.

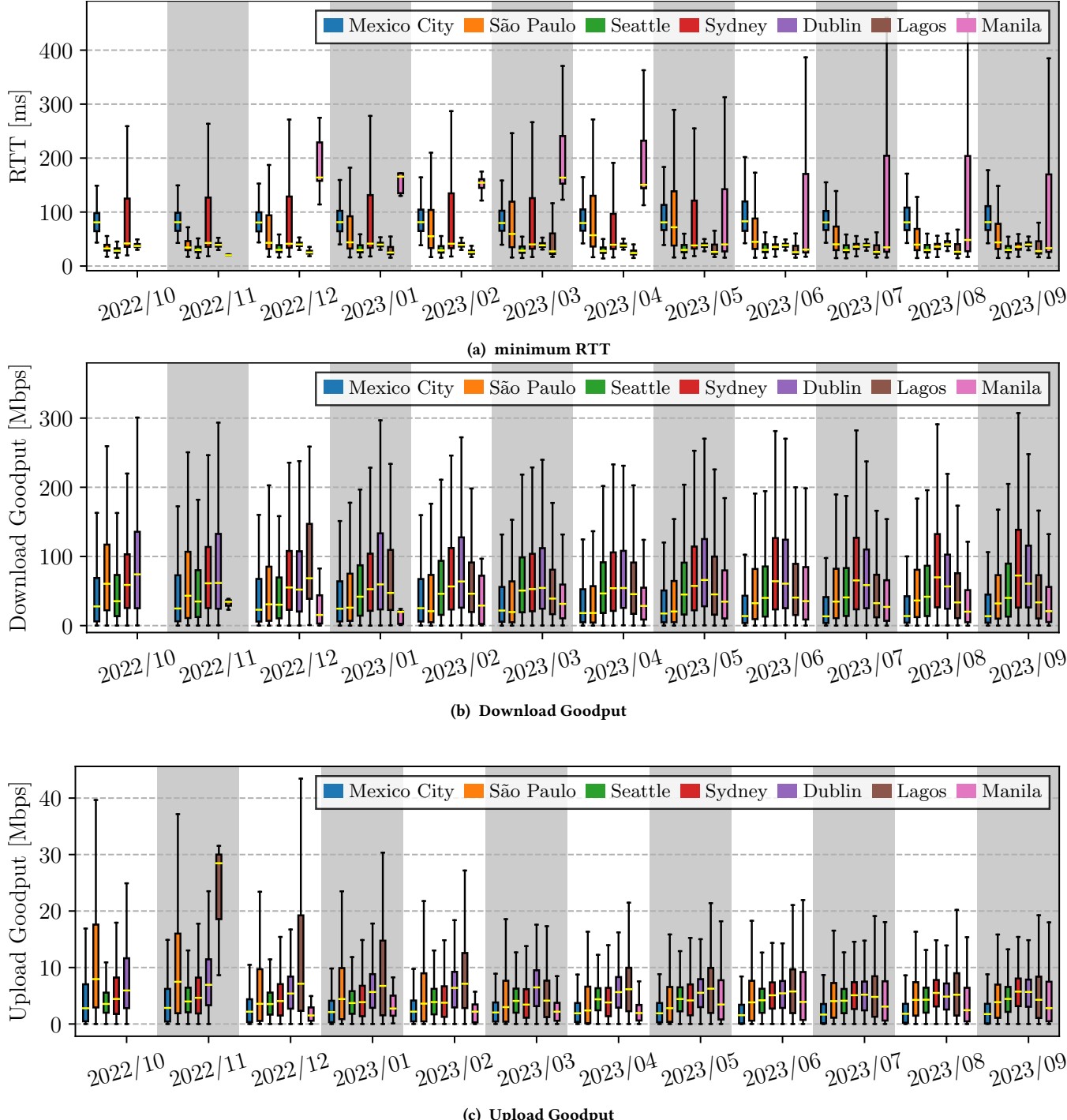

(a) minimum RTT

(b) Download Goodput

(c) Upload Goodput

Figure 23: Evolution of Starlink aggregate goodput ((a), (b)) and minimum RTT (c) during download measurements from cities in South America, North America, Europe, and Australia in the last 12 months.

| Country | Mobile Operator | ASN | AS Rank [5] |
|---|---|---|---|
| Australia | Telstra | AS1221 | 63 |
| | Optus | AS4804 | 4514 |
| | Vodafone | AS133612 | 7242 |
| Austria | A1 Telekom Austria | AS8447 | 152 |
| | T-Mobile Austria | AS8412 | 511 |
| | Magenta Telekom | AS25255 | 512 |
| Belgium | Proximus | AS5432 | 975 |
| | Telenet | AS6848 | 1231 |
| | Orange | AS47377 | 3458 |
| Brazil | TIM | AS26615 | 90 |
| | Claro | AS28573 | 7178 |
| | Vivo | AS18881 | 11754 |
| Canada | Bell | AS577 | 89 |
| | Telus | AS852 | 200 |
| | Rogers | AS812 | 241 |
| Chile | Movistar | AS7418 | 3893 |
| | Claro | AS27995 | 7275 |
| | Entel | AS27651 | 11848 |
| Colombia | Tigo Colombia | AS13489 | 1353 |
| | Claro | AS10620 | 11770 |
| | Colombia Móvil | AS27831 | 11775 |
| Czechia | Vodafone | AS16019 | 272 |
| | O2 Czech Republic | AS5610 | 828 |
| | T-Mobile | AS5588 | 2308 |
| Dominican Republic | Claro | AS6400 | 2636 |
| | Altice Dominicana | AS28118 | 3118 |
| | Tricom | AS27887 | 3539 |
| France | Orange | AS3215 | 204 |
| | SFR | AS15557 | 413 |
| | Bouygues Telecom | AS5410 | 1316 |
| Germany | Deutsche Telekom | AS3320 | 20 |
| | Vodafone | AS3209 | 237 |
| | O2 | AS6805 | 1917 |
| Greece | Vodafone Greece | AS3329 | 719 |
| | Forthnet GR | AS1241 | 915 |
| | Cosmote | AS29247 | 11995 |
| Guadeloupe | Dauphin Telecom | AS33392 | 6315 |
| Ireland | Eir | AS5466 | 1747 |
| | Vodafone | AS15502 | 3471 |
| | Three | AS13280 | 7364 |
| Italy | TIM | AS3269 | 244 |
| | Wind Tre | AS1267 | 434 |
| | Vodafone | AS30722 | 1049 |
| Japan | KDDI | AS2516 | 91 |
| | SoftBank | AS17676 | 111 |
| | NTT Docomo | AS9605 | 11764 |

| Country | Mobile Operator | ASN | AS Rank [5] |
|---|---|---|---|
| Kenya | Safaricom | AS33771 | 841 |
| | Airtel Kenya | AS36926 | 976 |
| | Telkom Kenya | AS12455 | 4566 |
| Martinique | Digicel | AS48252 | 3508 |
| Mexico | AT&T | AS28469 | 5558 |
| | Telcel | AS28403 | 15443 |
| Mozambique | Movitel | AS37342 | 12294 |
| | mCel | AS36945 | 21048 |
| Netherlands | KPN | AS1136 | 747 |
| | T-Mobile | AS50266 | 1268 |
| | Tele2 | AS13127 | 1413 |
| New Zealand | 2degrees | AS9790 | 250 |
| | Vodafone | AS9500 | 1924 |
| | Spark | AS4771 | 3913 |
| Nigeria | MTN | AS29465 | 885 |
| | Airtel | AS36873 | 1167 |
| | Glo | AS328309 | 23694 |
| Norway | Telenor | AS2119 | 255 |
| | NextGenTel AS | AS15659 | 4570 |
| | TELIA NORGE AS | AS12929 | 7314 |
| Peru | Movistar | AS6147 | 1735 |
| | Claro | AS12252 | 1741 |
| | Entel | AS21575 | 2276 |
| Philippines | Globe Telecom | AS132199 | 11910 |
| | Smart Communications | AS10139 | 12101 |
| Poland | Orange Polska | AS5617 | 136 |
| | T-Mobile Poland | AS12912 | 175 |
| Portugal | NOS | AS2860 | 809 |
| | MEO | AS15525 | 1329 |
| | Vodafone | AS12353 | 1592 |
| Puerto Rico | Claro | AS10396 | 1088 |
| | Liberty | AS14638 | 1242 |
| | T-Mobile | AS21928 | 5435 |
| Saint Barthélemy | Digicel | AS3215 | 204 |
| Spain | Orange | AS12479 | 332 |
| | Vodafone | AS12430 | 358 |
| | Movistar | AS3352 | 382 |
| Sweden | Tele2 Sweden | AS1257 | 195 |
| | Telia Company | AS3301 | 387 |
| | Telenor Sweden | AS8642 | 65529 |
| United Kingdom | O2 | AS5089 | 505 |
| | Vodafone | AS5378 | 5443 |
| | EE | AS12576 | 11745 |
| United States | T-Mobile | AS21928 | 5435 |
| | AT&T Mobility LLC | AS20057 | 7191 |
| | Verizon | AS22394 | 11784 |

Table 4: The selection of top-3 terrestrial MNOs (mobile network operators) for countries with Starlink M-Lab measurements.

