# OpenReview forum: "A Multifaceted Look at Starlink Performance"
_ACM.org/TheWebConf/2024/Conference — TheWebConf24 Oral_

### Official Review · Reviewer_uRUd · 2023-11-19

**Novelty:** 7
**Technical Quality:** 6

**Review:**

This paper performs a thorough study of starlink performance in the wild by capturing data over 34 countries which is the largest measurement of starlink performance analysis so far. Compared to the existing measurement papers on starlink performance, this paper extends the study significantly while providing several insights of the starlink network performance.

**Quality:**

The overall quality of the paper is good. The authors have provided the details about the methodology to collect the data and step-wise analysis of the data insights.

**Clarity:**

The writing of the paper is good. The presentation is clear and easy to follow and understand.

**Originality:**

In comparison to the existing literature, the work is original that extends the study of starlink performance over a large-scale analysis.

**Significance:**

The paper has significance with the web community; however, I feel that it has better connection with the measurement conferences (like ACM IMC).

Overall, I liked this paper and feel that it is of high quality. I would happy to recommend an acceptance for the paper. There are some clarifications that I have asked under the **Questions** field below; would be good if the authors can clarify the same.

**Questions:**

1) Is AS14593 the only AS that serve starlink clients? I am not an expert in this area, and so, just curious whether the measurement has substantial coverage of the starlink clients served.

2) How have you setup the zoom sessions? Are both the clients use starlink network? What is the video streaming performance if one client belongs to the high-speed terrestrial network, whereas another client connects starlink?

3) I am just curious, whether the competitive performance of starlink is due to the fact that it currently has much lower number of active clients compared to the terrestrial network, so it can support higher bandwidth? What would be the impact of larger number of active clients on the application performance?

**Reviewer Confidence:**

3: The reviewer is confident but not certain that the evaluation is correct

**Scope:**

3: The work is somewhat relevant to the Web and to the track, and is of narrow interest to a sub-community

---

### Official Review · Reviewer_bSfj · 2023-11-23

**Novelty:** 7
**Technical Quality:** 6

**Review:**

This is an interesting topic and an extremely thorough set of experiments. The authors explore the area with commendable thoroughness and the paper is easy to read. The measurements are extremely thorough and the authors should be commended.

A very minor thing but I found figure 5 quite hard to "work with" because the colour scale is arbitrary (green is between orange and red for example).  The scale is chosen in a way such that almost everything on the right hand map is blue (<40 ms).

The labelling of fig 6,7,8 is a little confusing as 6,8 explicitly state they are starlink measurements but 7 also is unless I badly misinterpret the text. A very minor point is the y-axis of these figures is given as a percentile but the scale runs to 1.0 not 100.

Unless I missed it you did not define "bent-pipe" performance (it's easy to look up but a sentence might help).

**Questions:**

While figure 11 does seem to show an FPS drop related to the reconfiguration interval it, in fact, appears to precede the reconfiguration. Is there a reason? Surely an FPS drop cannot be explained by a subsequent reconfiguration? This also seems to occur in figure 16. Is the reconfiguration interval actually slightly incorrect or is there some other explanation?

Presumably figuer 13 has an exclusion zone in the bottom right where it would not be physically possible for that RTT to occur over that physical distance (although obviously that is far from all of the physical distance).

Is bufferbloat the only possible explanation for the latency increase? (It's the only one that immediately comes to mind and the most obvious of course.)

**Reviewer Confidence:**

3: The reviewer is confident but not certain that the evaluation is correct

**Scope:**

4: The work is relevant to the Web and to the track, and is of broad interest to the community

---

### Official Review · Reviewer_4QWZ · 2023-11-23

**Novelty:** 5
**Technical Quality:** 6

**Review:**

## Summary

Low-earth orbit (LEO) constellations are emerging as a new consumer-facing access network option, especially in remote, or under-served areas. Starlink is one of the most popular such networks. This paper presents a comprehensive evaluation of the performance of Starlink. First, using M-Lab measurements, the paper compares Starlink to terrestrial cellular networks. Next, the paper evaluates the performance of Zoom and Amazon Luna over Starlink. Third, RIPE Atlas measurements are used to analyse the last-mile performance of the network. And finally, there is a deep-dive into Starlink's reconfiguration intervals.

## Reasons to accept
- The paper offers one of the most comprehensive measurement studies of Starlink performed to date.
- The analysis of the sources of delay and other effects is thorough.
- Discussion and evidence against the theory that the reconfiguration intervals are solely the result of satellite hand-offs is particularly interesting.

## Reasons to reject
- I don't have anything that really rises to the level of a reason to reject, but I have several author questions/points for clarification, which I expand below.

**Questions:**

- The comparison throughout the paper is with terrestrial cellular networks. Why was Zoom compared with the performance of a fixed-line network, and not using a 5G modem (as for Luna)?
- One limitation is the inability to split M-Lab results into those carried out over cellular networks. Eyeballing the list of ASes classified as "cellular networks", many of them carry a mix of mobile and fixed-line traffic. Is there anything further than can be done to explore this?

**Reviewer Confidence:**

2: The reviewer is willing to defend the evaluation, but it is likely that the reviewer did not understand parts of the paper

**Scope:**

3: The work is somewhat relevant to the Web and to the track, and is of narrow interest to a sub-community

---

### Official Review · Reviewer_XUiH · 2023-11-27

**Novelty:** 5
**Technical Quality:** 3

**Review:**

Strengths:
* A multi-modal analysis of Starlinks deployment.
* Significant amount of data is analyzed.

Weakness:
* The appendix is often required to understand some of the discussion/results.
* The takeaways generalize often with insufficient evidence: General claims are made without a lack of statistical rigor.

Details analysis:
* There is a potential bias and incorrectness from the geolocation tools: can you provide a discussion of the implications of these inaccuracies?
* The heatmap in figure 5 shows clear differences in colours -- this is at odds with the key takeaway #1. How do you reconcile these differences?
* The discussion of coverage and equivalence can be tested statistically. Equivalence by using distribution testing mechanisms. Coverage, the world can be broken into bing tiles (https://learn.microsoft.com/en-us/bingmaps/articles/bing-maps-tile-system) and the coverage can also be statistically compared.
* Figure 7 and figure 6 have different cities -- this. makes figure 8 a bit hard to contextualize.
* The 15 sec reconfiguration could lead to video degradation due to ABR depending on the buffering settings.  Such settings are ignored. For video, QoE should be measured more directly using application level metrics.
* The use-case analysis for Reunion island is a point example and not sufficient for broad generalizations in takeaway-#3.

Summary: This paper explores StarLinks along multiple dimensions and explores competitive from raw network metrics and application specific metrics. As a broad characterization, the paper does an ambitious analysis however it is often hard to understand because of reliance on the appendix. With regards to explore the competitiveness, this is often challenging as claims are made without statistical rigor (or underpinning) --- the takeaway often come across as unsupported.  I would like to encourage the authors to:
(1) reduce the scope to focus on the perhaps the first two takeaways which allows for more text and analysis without a heavy reliance on the appendix.
(2) Given the data, exploring statistical techniques and more rigorous comparison of existing data will help solidify the claims.

**Questions:**

(1) The paper makes broad claims about availability and 'competitiveness", how do you define competitiveness? how do the results substantiate this definition.
(2) There is clear bias in the data sources (e.g., M-Lab) and also fundamental differences in terms of coverage, how do you account for these nuanced but crucial differences?
(3) The analysis of real applications does not seem to mirror realistic deployment conditions, is there a reason why the focus on an idealized scenarios is crucial? This is clearly different from expectations in the field.

**Reviewer Confidence:**

3: The reviewer is confident but not certain that the evaluation is correct

**Scope:**

3: The work is somewhat relevant to the Web and to the track, and is of narrow interest to a sub-community

---

### Official Review · Reviewer_MMvf · 2023-11-30

**Novelty:** 6
**Technical Quality:** 5

**Review:**

The main aspects discussed in this article include:

- The latency and other network parameters of Starlink services across different regions
- A comparison of Starlink's capability of supporting real-time video conferencing & cloud gaming against existing network services
- An in-depth discussion of the latency introduced by bent-pipe mechanism
- A detailed analysis of Starlink's 15-second reconfiguration behavior

This paper stands out as one of the most comprehensive evaluation of the Starlink to date, with strengths as following:

- The testing sites are widespread, covering over 30 countries globally
- This paper goes beyond the application-layer evaluation to discuss and reveal the internal working of this black-box system

This paper is well-written, well-structured and easy to read, with sufficient data and robust argumentation.
However, I would suggest the authors consider the following for further improvement:

- Emphasize more on the discussion of bent-pipe and reconfiguration to highlight the novelty of the work
- When evaluating cloud gaming over Starlink, it may not be very appropriate to *execute in-game actions at pre-defined intervals*, as actions of users are randomly distributed

**Questions:**

The writing of the paper is very clear and I have nothing to question the author.

**Reviewer Confidence:**

4: The reviewer is certain that the evaluation is correct and very familiar with the relevant literature

**Scope:**

4: The work is relevant to the Web and to the track, and is of broad interest to the community

---

### Decision · Program_Chairs · 2024-01-22

**Decision:**

Accept (Oral)

**Comment:**

Reviewers broadly agree that this is a worthy contribution, with a wide geographic coverage and a significant amount of data analysed. The authors have also engaged thoroughly with the reviewers' comments and the majority of reviewers are satisfied with the proposed enhancements. The next version of the paper should incorporate all the proposed changes.